# A Population-Based Cohort Study of the Association between Visual Loss and Risk of Suicide and Mental Illness in Taiwan

**DOI:** 10.3390/healthcare11101462

**Published:** 2023-05-18

**Authors:** Chieh Sung, Chi-Hsiang Chung, Fu-Huang Lin, Wu-Chien Chien, Chien-An Sun, Chang-Huei Tsao, Chih-Erh Weng, Daphne Yih Ng

**Affiliations:** 1Graduate Institute of Medical Sciences, National Defense Medical Center, Taipei 11490, Taiwan; a652667@yahoo.com.tw; 2Department of Nursing, Keelung Chang Gung Memorial Hospital, Keelung 20401, Taiwan; w0672@cgmh.org.tw; 3School of Public Health, National Defense Medical Center, Taipei 11490, Taiwan; g694810042@gmail.com; 4Taiwanese Injury Prevention and Safety Promotion Association, Taipei 11490, Taiwan; 5Graduate Institute of Life Sciences, National Defense Medical Center, Taipei 11490, Taiwan; 6Department of Medical Research, Tri-Service General Hospital, National Defense Medical Center, Taipei 11490, Taiwan; changhuei@gmail.com; 7Department of Public Health, College of Medicine, Fu-Jen Catholic University, New Taipei City 24205, Taiwan; 040866@mail.fju.edu.tw; 8Big Data Research Center, College of Medicine, Fu-Jen Catholic University, New Taipei City 24205, Taiwan; 9Department of Microbiology and Immunology, Tri-Service General Hospital, National Defense Medical Center, Taipei 11490, Taiwan; 10Department of Family Medicine, Tri-Service General Hospital, National Defense Medical Center, Taipei 11490, Taiwan; conspicuous.com@gmail.com

**Keywords:** vision loss, National Health Insurance Research Database, retrospective cohort study, all-cause mortality, suicide, poor prognosis

## Abstract

The psychosocial and health consequences of ocular conditions that cause visual impairment (VI) are extensive and include impaired daily activities, social isolation, cognitive impairment, impaired functional status and functional decline, increased reliance on others, increased risk of motor vehicle accidents, falls and fractures, poor self-rated health, and depression. We aimed to determine whether VI increases the likelihood of a poor prognosis, including mental illness, suicide, and mortality over time. In this large, location, population-based, nested, cohort study, we used data from 2000 to 2015 in the Taiwan National Health Insurance Research Database (NHIRD), which includes diagnoses of all the patients with VI. Baseline features, comorbidities, and prognostic variables were evaluated using a 1:4-matched cohort analysis. Furthermore, comparisons were performed using Cox regression and Bonferroni-correction (for multiple comparisons) to study the association between VI and poor prognosis (mental illness, suicide). The study outcome was the cumulative incidence of poor prognosis among the visually impaired and controls. A two-tailed Bonferroni-corrected *p* < 0.001 was considered statistically significant. Among the 1,949,101 patients enlisted in the NHIRD, 271 had been diagnosed with VI. Risk factors for poor prognosis and the crude hazard ratio was 3.004 (95% confidence interval 2.135–4.121, *p* < 0.001). Participants with VI had an increased risk of poor prognosis according to the sensitivity analysis, with a poor prognosis within the first year and first five years. VI was associated with suicide and mental health risks. This study revealed that patients with VI have a nearly 3-fold higher risk of psychiatric disorders, including anxiety, depression, bipolar, and sleep disorders, than the general population. Early detection through comprehensive examinations based on increased awareness in the clinical context may help maintain visual function and avoid additional complications.

## 1. Introduction

Visual impairment (VI) is extremely common and affects more than 2.2 billion people worldwide [1]. Approximately 36 million people are blind, and an additional 217 million have significant (moderate-to-severe) VI [2]. According to Taiwan’s Ministry of the Interior, the number of visually impaired people in Taiwan has climbed from 19,423 in 1992 to 55,000 in 2022, with a 2000-person increase every year. The visually impaired population in Taiwan amounts to 0.24% of the overall population, compared to 0.5% in Europe and the United States; this is a low percentage in our location but is growing every year.

The wide-ranging psychosocial and health consequences of ocular conditions that cause VI include impaired activities of daily living, social isolation, cognitive impairment, impaired functional status and functional decline, increased reliance on others, risk of motor vehicle accidents, falls and fractures, poor self-rated health, and depression [3,4]. 

Adults with VI and disabling eye diseases have an increased risk of mortality [5] Moreover, case reports have suggested an association between VI and suicide. In a case-control study conducted in Sweden, researchers interviewed the families of 46 men and 39 women aged 65 years who completed suicide and compared the results with those of 84 men and 69 women aged 65 years who were still alive [6]. VI (odds ratio [OR] = 7.0; 95% confidence interval [CI]: 2.3–21.4), neurological disorders (OR = 3.8; 95% CI: 1.5–9.4), and malignant diseases (OR = 3.4; 95% CI: 1.2–9.8) were found to be independently associated with an increased risk of suicide. However, the statistical relevance of VI as a risk factor for suicide is debatable. VI has been identified as a risk factor in certain studies when combined with depression or poor overall health [7,8,9]. Age, male sex, disrupted marital status, mental and addictive disorders, depression, prior suicide attempts, family history of psychiatric disorders or suicide, a firearm in the house, and a recent severely stressful life event are all established risk factors for suicide [10]. However, in other investigations, VI did not demonstrate substantial hazard ratios or strong relationships with a higher risk of suicide [11,12,13,14].

Suicide and suicide attempts have been linked to chronic conditions such as cancer, and the number of health conditions increases the risk of suicide attempts [15]. Therefore, the risk of suicide may be influenced both directly and indirectly by VI. For example, a link between VI and self-rated health, which has been shown to be a predictor of mortality, has been discovered [16,17,18]. Thus, this study aimed to determine whether VI increases the risk of mental illness and/or suicide in a representative Taiwanese population.

## 2. Materials and Methods

### 2.1. Data Source

The Taiwan National Health Insurance (NHI) program was established in 1995 and covers >99% of the Taiwanese population (>23 million beneficiaries). The Taiwan NHI Research Database (NHIRD) contains the following encrypted data: patient identification number; date of birth; sex; date of admission and discharge; International Classification of Diseases, Ninth Revision, Clinical Modification (ICD−9-CM) diagnostic and procedure codes (up to five each); and outcomes. The Longitudinal Health Insurance Database (LHID) 2005, a subset of the NHIRD, has been used in this study. It contains data on the medical service utilization of approximately 1 million randomly selected beneficiaries, who represented approximately 5% of the Taiwanese population in 2005. The NHIRD was used to extract data from 2000 to 2015. The NHI administration regularly conducts random reviews of medical records to ensure diagnostic accuracy.

This study was a population-based cohort tracking study that recruited outpatients and inpatients from the Taiwan Longitudinal Health Insurance Database (LHID) between 1 January 2000 and 31 December 2015. Tandem Inpatient expenditures by admissions (DD), Registry for contracted medical facilities (HOSB), Registry for beneficiaries (ID), Registry for catastrophic illness patients (HV), variables include diagnosis, surgery, disposition, hospitalization and discharge dates, length of stay and medical costs; Registry for contracted medical facilities (HOSB)the variables include hospital location and hospital level.

This study was conducted in accordance with the Declaration of Helsinki. The institutional review board of Tri-Service General Hospital at the National Defense Medical Center in Taipei, Taiwan, approved this study, and the requirement for individual consent was waived because all identifying data were encrypted (TSGHIRB No. E202216004). The NHIRD is a freely accessible database that contains de-identified patient information to protect patient anonymity.

### 2.2. Study Design and Participants

This study is a retrospective cohort study that uses secondary database analysis. According to the International Classification of Diseases (ICD) and Related Health Problems, the World Health Organization divides VI into five categories. The first and second categories are moderate or severe VI with excellent visual acuity equal to or better than 3/60 (0.05). The third category is blindness with visual acuity between 3/60 (0.05) to 1/60 (0.02); the fourth category is blindness with visual acuity between 1/60 (0.02) to light perception; and the fifth category is blindness with no light perception. This study included a cohort of patients from the LHID database who were newly diagnosed with VI (ICD-9-CM 369.3 and 369.4). We excluded patients diagnosed with VI before 2000, poor prognosis before VI, unknown sex, and incomplete tracking data. The inclusion and exclusion criteria are shown in Figure 1. Moreover, the date of the diagnosis of VI was used as the index date. Participants in the control group were selected from the LHID 2005 cohort. The study and control cohorts were matched 1:4 according to sex, age, and the index date.

### 2.3. Outcome Measurement and Comorbidities

We individually assessed medical comorbidities associated with VI at enrollment and during the entire follow-up period, including diabetes mellitus (ICD-9-CM 250), hypertension (HTN; ICD-9-CM 401-405), renal disease (ICD-9-CM 580-589), hyperlipidemia (ICD-9-CM 272), thyrotoxicosis (ICD-9-CM 242), septicemia (ICD-9-CM 003.1, 036.1, 038), pneumonia (ICD-9-CM 480-486), chronic liver disease (ICD-9-CM 571), injury (ICD-9-CM 800-999), and tumor (ICD-9-CM 140-208); these are listed in Appendix A.

All participants were followed up from the index date until the first diagnosis of VI, poor prognosis, death, withdrawal from the NHI program, or 31 December 2015. The covariates included sex, age group, geographical area of residence (north, center, south, and east of Taiwan), urbanization level of residence (levels 1–4), and monthly income (in New Taiwan Dollars: <18,000, 18,000–34,999, and ≥35,000). The urbanization level of residence was defined according to the population and various indicators of development. Level 1 was defined as a population of >1,250,000 with a specific designation of political, economic, cultural, and metropolitan development. Level 2 was defined as a population between 500,000 and 1,249,999, with an important role in politics, the economy, and culture. Finally, urbanization levels 3 and 4 were defined as populations between 149,999 and 499,999 and <149,999, respectively.

Risk factors evaluated for poor prognosis included mental disorders, at least three outpatient or inpatient visits, anxiety disorders (ICD−9-CM 300), depression (ICD−9-CM 296.2, 296.3, 300.4, and 311), bipolar disorder (ICD−9-CM 296.0, 296.4–296.8), sleep disorders (ICD−9-CM 307.4 and 780.5), post-traumatic stress disorder/acute stress disorder (ICD−9-CM 308 and 309.81), dementia (ICD−9-CM 290.0–290.4, 290.8, 290.9, and 331.0), eating disorders (ICD-CM 307.1 and 307.5), substance-related disorders (SRD; ICD-CM 291–292, 303.3, 303.9, and 304–305), psychotic disorders (ICD-CM 295 and 297–298), autism (ICD-CM 299.00), other mental disorders (not listed above, ICD−9-CM 290–319), suicide (ICD−9-CM E950–E959), death from all causes (ICD−9-CM 800–999), suicide mortality (ICD−9-CM E950–E959, immediate and subsequent mortality), and non-suicide mortality (ICD−9-CM000–999, E800–E949, E960–E969); these are listed in Appendix A.

### 2.4. Statistical Analyses

To investigate the association between VI and risk of suicide and mental illness, we conducted the following statistical analyses to compare the clinical characteristics of the participants in the case and control groups. The clinical characteristics of the participants are expressed numerically. We compared the distribution of categorical characteristics and baseline comorbidities between the case and control groups using Fisher’s exact test and the chi-squared test. Continuous variables are presented as means and standard deviations and compared using the *t*-test.

As the primary goal of this study was to determine whether the clinical characteristics of patients are associated with poor prognosis, the Cox regression analyses were used to determine the risk of poor prognosis; the results are presented as hazard ratios (HRs) with the associated 95% CIs.

Associations between time-to-event outcomes and clinical characteristics were examined using the Kaplan–Meier method and multivariate Cox regression analysis with stepwise selection; the results are presented as adjusted HRs with the corresponding 95% CIs. The poor prognosis incidence (per 10^5^ person-years) was calculated based on sex, age, and comorbidities for each cohort. Adjustments were made for age, sex, and concomitant comorbidities for inclusion in the multivariate model.

Bonferroni-correction for multiple comparisons was applied. A two-tailed Bonferroni-corrected *p* < 0.001 was considered statistically significant. All statistical analyses were performed using IBM SPSS Statistics for Windows version 22.0 (released 2013, IBM Corp., Armonk, NY, USA).

## 3. Results

Among the 1,949,101 patients in the LHID 2005 from the NHIRD, 539 had been diagnosed with VI. In total, 271 patients were assigned to the study cohort, and 1084 age-, sex-, and comorbidity-matched patients were assigned to the comparison cohort (Figure 1). The baseline data of the patients and control groups are shown in Table 1. The average age of the VI cohort was 39.01 ± 16.83 years, and the proportion of male patients was 52.03%. Among the study population, the majority of patients were aged ≦19 years (36.53%); 27.31% of patients were 65 years and older, 20.66% were 20–44 years, and 15.50% were 45−64 years old. Our findings show no significant differences in sex, age, thyrotoxicosis, septicemia, tumor, or season between the groups with and without VI after matching.

Furthermore, we analyzed individual outcomes, such as dementia, eating disorders, SRD, psychotic disorders, autism, and other mental disorders (Table 2). In general, VI was associated with an increased risk of individual outcomes, including dementia, eating disorders, SRD, and other mental disorders, even after excluding individuals with poor prognoses within the first year and first five years. Moreover, VI was associated with an increased risk of anxiety, depression, bipolar disorder, and sleep disorders, as well as individual types of all-cause mortality.

Figure 2 shows the Kaplan–Meier survival curve of patients with poor prognosis stratified by visual loss using the log-rank test; patients with visual loss had a significantly higher cumulative risk of developing a poor prognosis 16 years after the index date (log-rank test, *p* < 0.001).

Table 3 shows the Cox regression analysis of the factors associated with the risk of poor prognosis. The crude HR was 3.004 (95% CI: 2.135–4.121, *p* < 0.001). After adjusting for sex, age group, geographical area of residence, urbanization level of residence area, and monthly income, the adjusted HR was 2.956 (95% CI: 1.984–3.960, *p* < 0.001). Age, urbanization level, and level of care correlated with poor prognosis. Moreover, poor prognosis tended to occur in patients older than 45 years, regardless of urbanization level and level of care (*p* < 0.05). Notably, this study included more males than females. Moreover, the level of urbanization increased with the frequency of chronic diseases (e.g., HTN, renal disease), which more likely resulted in the crude and adjusted HR of poor prognosis (*p* < 0.05).

The patients were stratified by the variables presented in Table 3, and adjusted hazard ratios of different subgroups were calculated (Table 4). The visual loss group encountered 49 medical events due to first diagnosed poor prognosis in the 2552.73 person-years (PY) observed, representing a rate of 936.93 per 105 PYs; the group without visual loss encountered 97 medical events in the 10,352.96 person-years (PY) observed, representing a rate of 1614 per 105 PYs. After Bonferroni-correction for multiple comparisons, when compared to those without visual loss, patients with visual loss poor prognosis showed patients with visual loss poor prognosis ratio of 2.956 (95% CI 1.984–3.960, *p* < 0.001).

This study was designed to analyze the short-, medium-, and long-term effects of VI on patients. Factors in the poor prognosis subgroups (overall, 1-year, and 5-year subgroups) posed a significant risk compared to the group without VI (Table 5). Visual loss was associated with an increased risk of poor prognosis (aHR, 2.956; 95% CI, 1.984–3.960; *p* < 0.001); post-traumatic stress disorder (PTSD)/acute stress disorder (ASD) in VI was 5.835 fold higher (95% CI, 3.866−7.892; *p* < 0.001) than those without VI. In the present study, visual loss was associated with an increased risk of any poor prognosis from the subgroups, as well as with an increased risk of specific types of personality disorders, including mental disorders, anxiety, depression, bipolar, sleep disorders, PTSD/ASD, dementia, eating disorders, SRD, psychotic disorders, autism, other mental disorders, and suicide (Table 5). After Bonferroni-correction for multiple comparisons, notable increases in the first year in SRD with VI was 5.745-fold (95% CI, 3.882–7.759; *p* < 0.001), PTSD/ASD was 5.526-fold (95% CI, 3.884–7.801; *p* < 0.001), suicide mortality was 2.402-fold (95% CI, 1.362–3.780; *p* < 0.001), and non-suicide mortality was 3.765-fold (95% CI, 2.087–7.184; *p* < 0.001). After the fifth year, the increase in depression was 5.219-fold (95% CI, 3.467–6.920; *p* < 0.001). These associations remained significant after the visual loss diagnoses in the first year, persisting even after the first five years after experiences with visual loss, according to the sensitivity analysis (Table 5).

## 4. Discussion

People with VI are at a higher risk of poor mental health outcomes, as well as physical comorbidities [19,20,21]. According to a study published in 2000, two-thirds (66.6%) of patients admitted for inpatient treatment for depression showed a diminished perception of ambient light [22]. According to a cross-sectional study comprising 213 participants (with depression and without depression), patients with depression were 4.5 times more likely to report lower perception of ambient light from age-related eye illness than those without depression. However, the mechanism underlying the association between depression and impaired perception remains unknown [23].

Previous studies have shown an association between VI and self-rated health [16,24,25]. Reduced vision had an independent influence on global health ranking by those under the age of 80 years [24]. Participants in the study who reported VI were twice as likely as those who did not report VI to indicate poor self-rated health (OR = 2.13, 95% CI: 1.94–2.33). Self-reported health is thought to reflect physical health conditions. When reporting self-rated health, a respondent may also consider healthy or unhealthy habits and activities [23].

Previous studies have reported that increased exposure to potentially stressful situations is associated with an increased risk of mental health problems in persons with VI. Studies have linked visual impairments to a high prevalence of post-traumatic stress disorder [25], higher risk of depression and anxiety [26,27], and burdensome life experiences such as loneliness [28]. While it has been discovered that mental health issues are more common in young persons with VI than in older persons with VI [29], studies have also indicated that older adults with VI have a higher prevalence of a range of mental health disorders when compared with similarly aged people in the general population [30]. Therefore, people with VI are more likely to have mental health problems, regardless of age; this is in line with the results of our study.

The present study found that, in addition to depression, VI may directly increase the risk of suicide (HR: 1.50, 95% CI: 0.90–2.49). When non-ocular characteristics, including medical comorbidities and self-rated health, were considered, individuals with VI had a 64% (HR: 1.64, 95% CI: 0.99–2.72) greater risk of death by suicide [31]. While blindness due to age-related illnesses (e.g., age-related macular degeneration), diabetic retinopathy, and glaucoma is not reversible, many impairments may be cured or prevented entirely, which may prevent development of several mental health disorders [32]. Nevertheless, despite their frequent contact with patients and understanding of the devastating effects of VI, ophthalmologists seldom diagnose or treat depression [33]. This decision making involves ophthalmologists’ ability to diagnose depression and suicidal behavior, as well as knowing when to send patients for psychiatric examination and care. Further research is needed to determine the frequency of cases with psychological difficulties that ophthalmologists encounter and appropriately manage. In addition, efforts to teach ophthalmologists in residency to effectively manage suicidal behaviors caused by VI are warranted.

PTSD may follow an exceptionally threatening or horrifying event, where the person experiencing it feels a severe threat of injury or death. Common symptoms of PTSD are re-experiencing the event in the form of flashbacks or nightmares, avoidance of stimuli associated with the event, alterations in cognition and mood, and increased arousal and reactivity [34]. As identified in a recent review, the one study that assessed PTSD prevalence specifically in people with VI was concerned with adolescents in a war conflict area [35]. This study found a lower prevalence of PTSD among those with impaired vision or hearing compared with those without impairments (4.2% versus 11.4%), which was explained by a lower exposure to traumatic events among those with VI [36]. However, in previous reviews, the prevalence estimates of PTSD in populations prone to VIs (older people, primary care patients) have ranged from 1.7% to 32.5% [37,38], which is both lower and higher, respectively, than those found in general population samples [39,40]. Thus, more studies are warranted to conclude whether individuals with VI are at a higher risk of PTSD.

While the current state of knowledge suggests that people with a visual impairment are more vulnerable to potentially traumatic events, few studies have examined the prevalence of PTSD. PTSD can occur after an exceptionally threatening or horrifying event in which the person experiencing it felt a severe threat of injury or death. Common symptoms of PTSD include re-experiencing the event in the form of flashbacks or nightmares, avoidance of stimuli associated with the event, and altered perceptions of the event [34].

According to a recent analysis, the one study that explicitly investigated PTSD prevalence in persons with a visual impairment was concerned with teenagers in a military conflict region [35]. This study revealed a lower prevalence of PTSD among people with impaired vision or hearing compared to those without impairments (4.2% versus 11.4%) [36], which was explained by the impaired group’s lesser exposure to stressful situations. PTSD prevalence estimates in populations prone to visual impairments (older persons, primary care patients) have ranged from 1.7% to 32.5% [37,38], which is both greater and lower than that seen in general population samples [39,40]. More study is needed to determine whether those with visual impairments are more likely to develop PTSD. More study is needed to determine which life experiences may be to blame for the probable difference in PTSD prevalence between the VI population and the general population.

Synthesis the above studies, PTSD is prone to forming in people with VI. Nonetheless, the degree and impact of PTSD in this population remains uncertain, and further investigation is necessary to advance knowledge regarding these aspects. This would entail conducting larger comparative studies using dependable methods and valid assessment tools. The current diagnostic instruments for traumatic events and PTSD must be validated and, if necessary, modified for individuals with VI. First, the influence of VI in the manifestation of PTSD symptoms, particularly intrusions, avoidance, and hyperarousal, must be considered. Second, PTSD and traumatic brain injury often co-occur, particularly in ex-service personnel, and traumatic brain injury may result in VI. Consequently, it may become challenging to attribute a symptom to a specific diagnosis. Third, PTSD diagnosis may be associated with an increase in vision problems due to heightened awareness and reporting of vision problems, neurophysiological manifestations, and medication side effects. Mental health professionals, vision rehabilitation specialists, and eye care providers should be acquainted with these factors to improve identification and treatment of PTSD in this population.

The major strength of our study is its population-based design. However, this study has some limitations. First, the NHIRD did not annually assess important risk factors associated with disabling eye conditions such as smoking, a risk factor for cataracts, and age-related maculopathy. Second, the NHIRD does not provide detailed information on variables, including socioeconomic factors, occupation, unhealthy behaviors, amount of alcohol consumption, and the genetic background of the subjects, that may affect the association between VI and poor mental health outcomes. Lastly, the study participants were selected based on their medical records in the NHIRD, so our ICD−9-CM code does not identify congenital VI and does not include an assessment of some risk factors associated with suicide, such as depression. Other than that, the fact that not all suicides are reported as suicides, may underestimate the data between VI and suicides. The bias caused by unknown confounders could not be avoided in this retrospective cohort study despite meticulous adjustments. Nevertheless, multivariate logistic regression models and Bonferroni-correction for multiple comparisons were used to adjust our results. The recommendation for future studies with higher methodological rigor is appropriate to address the limitations of the current study.

## 5. Conclusions

This study revealed that patients with VI have a nearly three-fold higher risk of mental disorders, including anxiety, depressive, bipolar, and sleep disorders, than the general population. After the sensitivity test, excluding the first year, patients with VI were found to have a nearly five-fold higher risk for PTSD/ASD. When the first five years were excluded, the data showed patients with VI to have a nearly five-fold higher risk for depression. Traumatic experiences appear to have a significant impact on the mental health in people with VI, and these results highlight their need for mental health care. The high prevalence of PTSD lends credence to the suggestion for a better-adapted health-care system for persons with visual impairment. People who are visually impaired may have a greater threshold for asking for assistance. Equally significant, health workers are unaware of the mental challenges connected with visual impairments [41].

## Figures and Tables

**Figure 1 healthcare-11-01462-f001:**
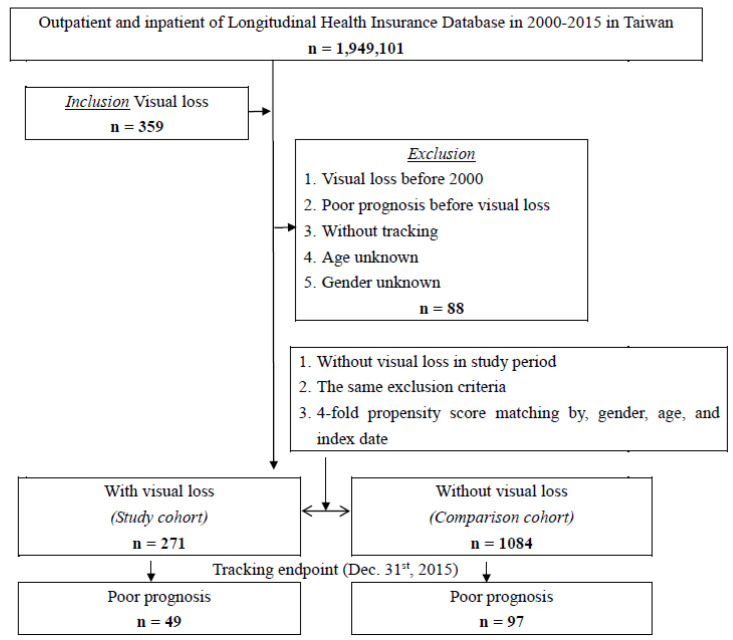
Flowchart of the study sample selection from the National Health Insurance Research Database in Taiwan.

**Figure 2 healthcare-11-01462-f002:**
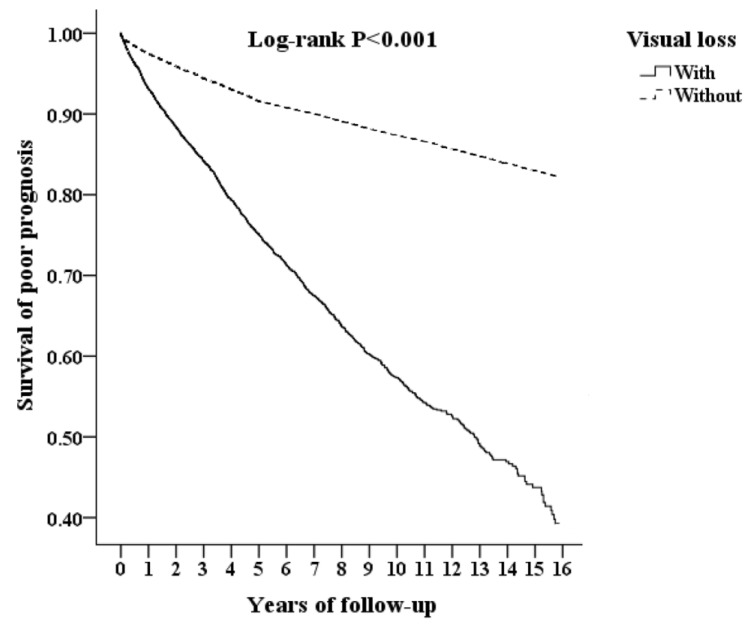
Kaplan–Meier for survival of poor prognosis stratified by visual loss with log-rank test.

**Table 1 healthcare-11-01462-t001:** Baseline characteristics of the participants.

Visual Loss	Total		With		Without		*p*
Variables	n	%	n	%	n	%	
Total	1355		271	20.00	1084	80.00	
Gender							0.999
Male	705	52.03	141	52.03	564	52.03	
Female	650	47.97	130	47.97	520	47.97	
Age (years)	39.12 ± 16.97		39.01 ± 16.83		39.15 ± 17.01		0.822
Age groups (yrs)							0.999
≦19	495	36.53	99	36.53	396	36.53	
20−44	280	20.66	56	20.66	224	20.66	
45−64	210	15.50	42	15.50	168	15.50	
≧65	370	27.31	74	27.31	296	27.31	
Insured premium (NT$)							<0.001
<18,000	1085	80.07	212	78.23	873	80.54	
18,000–34,999	149	11.00	37	13.65	112	10.33	
≧35,000	121	8.93	22	8.12	99	9.13	
DM							<0.001
Without	1189	87.75	226	83.39	963	88.84	
With	166	12.25	45	16.61	121	11.16	
HTN							<0.001
Without	1190	87.82	234	86.35	956	88.19	
With	165	12.18	37	13.65	128	11.81	
Renal disease							<0.001
Without	1183	87.31	229	84.50	954	88.01	
With	172	12.69	42	15.50	130	11.99	
Hyperlipidemia							<0.001
Without	1270	93.73	249	91.88	1021	94.19	
With	85	6.27	22	8.12	63	5.81	
Thyrotoxicosis							0.862
Without	1288	95.06	256	94.46	1032	95.20	
With	67	4.94	15	5.54	52	4.80	
Septicemia							0.999
Without	1340	98.89	268	98.89	1072	98.89	
With	15	1.11	3	1.11	12	1.11	
Pneumonia							0.001
Without	1230	90.77	243	89.67	987	91.05	
With	125	9.23	28	10.33	97	8.95	
CLD							0.014
Without	1227	90.55	243	89.67	984	90.77	
With	128	9.45	28	10.33	100	9.23	
Injury							<0.001
Without	1173	86.57	215	79.34	958	88.38	
With	182	13.43	56	20.66	126	11.62	
Tumor							0.972
Without	1319	97.34	264	97.42	1055	97.32	
With	36	2.66	7	2.58	29	2.68	
Season							0.999
Spring (Mar–May)	325	23.99	65	23.99	260	23.99	
Summer (Jun–Aug)	330	24.35	66	24.35	264	24.35	
Autumn (Sep–Nov)	355	26.20	71	26.20	284	26.20	
Winter (Dec–Feb)	345	25.46	69	25.46	276	25.46	
Location							<0.001
Northern Taiwan	383	28.27	78	28.78	305	28.14	
Central Taiwan	375	27.68	72	26.57	303	27.95	
Southern Taiwan	375	27.68	76	28.04	299	27.58	
Eastern Taiwan	152	11.22	32	11.81	120	11.07	
Outlying islands	70	5.17	13	4.80	57	5.26	
Urbanization level							<0.001
1 (The highest)	387	28.56	78	28.78	309	28.51	
2	405	29.89	81	29.89	324	29.89	
3	270	19.93	49	18.08	221	20.39	
4 (The lowest)	293	21.62	63	23.25	230	21.22	
Level of care							<0.001
Hospital center	415	30.63	100	36.90	315	29.06	
Regional hospital	511	37.71	94	34.69	417	38.47	
Local hospital	429	31.66	77	28.41	352	32.47	

*p*: Chi-square/Fisher exact test on category variables and *t*-test on continue variables, DM: diabetes mellitus; HTN: hypertension; CLD: chronic liver disease.

**Table 2 healthcare-11-01462-t002:** Study endpoint characteristics of the participants.

Visual loss	Total		With		Without		*p*
Variables	n	%	n	%	n	%	
Total	1355		271	20.00	1084	80.00	
Poor prognosis							<0.001
Without	1209	89.23	222	81.92	987	91.05	
With	146	10.77	49	18.08	97	8.95	
Gender							0.999
Male	705	52.03	141	52.03	564	52.03	
Female	650	47.97	130	47.97	520	47.97	
Age (yrs)	48.70 ± 18.80		47.85 ± 18.22		48.91 ± 18.94		0.003
Age groups (yrs)							0.012
≦19	480	35.42	97	35.79	383	35.33	
20−44	272	20.07	53	19.56	219	20.20	
45−64	191	14.10	41	15.13	150	13.84	
≧65	412	30.41	80	29.52	332	30.63	
Insured premium (NT$)							<0.001
<18,000	1085	80.07	212	78.23	873	80.54	
18,000−34,999	149	11.00	37	13.65	112	10.33	
≧35,000	121	8.93	22	8.12	99	9.13	
DM							<0.001
Without	1183	87.31	222	81.92	961	88.65	
With	172	12.69	49	18.08	123	11.35	
HTN							<0.001
Without	1186	87.53	232	85.61	954	88.01	
With	169	12.47	39	14.39	130	11.99	
Renal disease							<0.001
Without	1178	86.94	226	83.39	952	87.82	
With	177	13.06	45	16.61	132	12.18	
Hyperlipidemia							<0.001
Without	1260	92.99	243	89.67	1017	93.82	
With	95	7.01	28	10.33	67	6.18	
Thyrotoxicosis							<0.001
Without	1278	94.32	252	92.99	1026	94.65	
With	77	5.68	19	7.01	58	5.35	
Septicemia							0.972
Without	1334	98.45	267	98.52	1067	98.43	
With	21	1.55	4	1.48	17	1.57	
Pneumonia							0.447
Without	1225	90.41	240	88.56	985	90.87	
With	130	9.59	31	11.44	99	9.13	
CLD							0.285
Without	1218	89.89	235	86.72	983	90.68	
With	137	10.11	36	13.28	101	9.32	
Injury							<0.001
Without	1165	85.98	211	77.86	954	88.01	
With	190	14.02	60	22.14	130	11.99	
Tumor							0.943
Without	1316	97.12	263	97.05	1053	97.14	
With	39	2.88	8	2.95	31	2.86	
Season							0.002
Spring	317	23.39	62	22.88	255	23.52	
Summer	351	25.90	73	26.94	278	25.65	
Autumn	340	25.09	69	25.46	271	25.00	
Winter	347	25.61	67	24.72	280	25.83	
Location							<0.001
Northern Taiwan	386	28.49	77	28.41	309	28.51	
Central Taiwan	373	27.53	72	26.57	301	27.77	
Southern Taiwan	377	27.82	77	28.41	300	27.68	
Eastern Taiwan	157	11.59	33	12.18	124	11.44	
Outlying islands	62	4.58	12	4.43	50	4.61	
Urbanization level							<0.001
1 (The highest)	383	28.27	78	28.78	305	28.14	
2	410	30.26	83	30.63	327	30.17	
3	269	19.85	45	16.61	224	20.66	
4 (The lowest)	293	21.62	65	23.99	228	21.03	
Level of care							<0.001
Hospital center	409	30.18	99	36.53	310	28.60	
Regional hospital	514	37.93	93	34.32	421	38.84	
Local hospital	432	31.88	79	29.15	353	32.56	

*p:* Chi-square/Fisher exact test on category variables and *t*-test on continue variables, DM: diabetes mellitus; HTN: hypertension; CLD: chronic liver disease.

**Table 3 healthcare-11-01462-t003:** Factors of poor prognosis according to Cox regression.

Variables	Crude HR	95% CI	95% CI	*p*	aHR	95% CI	95% CI	*p*
Visual Loss								
Without	Reference				Reference			
With	3.004	2.135	4.121	<0.001	2.956	1.984	3.960	<0.001
Gender								
Male	1.454	1.112	1.893	<0.001	1.352	1.089	1.798	0.001
Female	Reference				Reference			
Age groups (yrs)								
≦19	Reference				Reference			
20−44	1.125	1.044	2.030	0.007	1.086	0.972	1.972	0.083
45−64	1.597	1.125	2.224	<0.001	1.459	1.033	2.106	0.016
≧65	1.886	1.301	2.482	<0.001	1.767	1.255	2.323	<0.001
Insured premium (NT$)								
<18,000	Reference				Reference			
18,000−34,999	0.764	0.489	1.149	0.589	0.873	0.617	1.301	0.677
≧35,000	0.688	0.375	0.952	0.002	0.770	0.486	1.095	0.124
DM								
Without	Reference				Reference			
With	2.698	1.798	3.995	<0.001	2.555	1.726	3.870	<0.001
HTN								
Without	Reference				Reference			
With	2.485	1.588	3.372	<0.001	2.304	1.448	3.271	<0.001
Renal disease								
Without	Reference				Reference			
With	2.656	1.672	3.501	<0.001	2.612	1.562	3.399	<0.001
Hyperlipidemia								
Without	Reference				Reference			
With	1.652	1.153	1.983	<0.001	1.583	1.097	1.876	0.004
Thyrotoxicosis								
Without	Reference				Reference			
With	1.108	0.865	1.404	0.186	1.085	0.722	1.315	0.274
Septicemia								
Without	Reference				Reference			
With	2.892	1.986	3.971	<0.001	2.506	1.872	3.501	<0.001
Pneumonia								
Without	Reference				Reference			
With	1.682	1.135	2.543	<0.001	1.446	1.026	2.489	0.028
CLD								
Without	Reference				Reference			
With	1.897	1.304	2.703	<0.001	1.587	1.104	2.555	<0.001
Injury								
Without	Reference				Reference			
With	2.359	1.683	2.979	<0.001	2.184	1.597	2.862	<0.001
Tumor								
Without	Reference				Reference			
With	1.906	1.401	2.345	<0.001	1.886	1.358	2.270	<0.001
Season								
Spring	Reference				Reference			
Summer	1.245	0.784	1.771	0.266	1.098	0.567	1.505	0.489
Autumn	1.579	0.896	1.896	0.193	1.225	0.704	1.722	0.276
Winter	1.606	1.001	2.035	0.050	1.297	0.779	1.781	0.251
Location					Multicollinearity with urbanization level			
Northern Taiwan	Reference				Multicollinearity with urbanization level			
Central Taiwan	0.909	0.539	1.212	0.471	Multicollinearity with urbanization level			
Southern Taiwan	0.976	0.571	1.277	0.433	Multicollinearity with urbanization level			
Eastern Taiwan	0.733	0.452	1.083	0.592	Multicollinearity with urbanization level			
Outlying islands	0.562	0.233	0.864	<0.001	Multicollinearity with urbanization level			
Urbanization level								
1 (The highest)	1.896	1.390	2.286	<0.001	1.572	1.137	2.030	<0.001
2	1.834	1.377	2.207	<0.001	1.510	1.108	1.997	<0.001
3	1.567	1.259	1.995	<0.001	1.372	1.024	1.866	0.027
4 (The lowest)	Reference				Reference			
Level of care								
Hospital center	2.862	2.030	3.782	<0.001	2.385	1.884	2.977	<0.001
Regional hospital	2.134	1.771	2.853	<0.001	1.786	1.275	2.386	<0.001
Local hospital	Reference				Reference			

HR = hazard ratio, CI = confidence interval, aHR = Adjusted HR: Adjusted variables listed in the table, DM: diabetes mellitus; HTN: hypertension; CLD: chronic liver disease.

**Table 4 healthcare-11-01462-t004:** Factors of poor prognosis stratified by variables listed in the table by using Cox regression and Bonferroni correction for multiple comparisons.

Visual Loss	With			Without *(Reference)*			With vs. Without *(Reference)*			
Strarified	Events	PYs	Rate (per 10^5^ PYs)	Events	PYs	Rate (per 10^5^ PYs)	aHR	95% CI	95% CI	*p*
Total	49	2552.73	1919.51	97	10,352.96	936.93	2.956	1.984	3.960	< 0.001
Gender										
Male	26	1328.84	1956.59	51	5386.61	946.79	2.943	1.975	3.942	< 0.001
Female	23	1223.89	1879.25	46	4966.35	926.23	2.889	1.939	3.871	< 0.001
Age groups (yrs)										
≦19	16	913.25	1751.98	33	3657.91	902.15	2.766	1.857	3.705	< 0.001
20−44	9	499.61	1801.41	19	2091.24	908.55	2.824	1.895	3.782	< 0.001
45−64	8	386.35	2070.66	14	1432.60	977.24	3.017	2.025	4.043	< 0.001
≧65	16	753.52	2123.37	31	3171.21	977.54	3.093	2.077	4.144	< 0.001
Insured premium (NT$)										
<18,000	40	1996.95	2003.05	78	8337.25	935.56	3.049	2.046	4.084	< 0.001
18,000−34,999	7	353.11	1982.39	10	1068.84	935.59	3.017	2.025	4.043	< 0.001
≧35,000	2	202.67	986.83	9	946.87	950.50	1.479	0.992	1.981	0.057
DM										
Without	39	2091.18	1864.98	86	9179.37	936.88	2.835	1.903	3.798	< 0.001
With	10	461.55	2166.61	11	1173.59	937.29	3.292	2.210	4.410	< 0.001
HTN										
Without	40	2185.00	1830.66	84	9111.44	921.92	2.828	1.898	3.788	< 0.001
With	9	367.73	2447.45	13	1241.52	1047.10	3.328	2.235	4.459	< 0.001
Renal disease										
Without	39	2128.85	1831.98	83	9092.35	912.86	2.858	1.918	3.829	< 0.001
With	10	423.88	2359.16	14	1260.61	1110.57	3.025	2.030	4.053	< 0.001
Hyperlipidemia										
Without	43	2288.98	1878.57	91	9713.08	936.88	2.855	1.917	3.826	< 0.001
With	6	263.75	2274.88	6	639.88	937.68	3.455	2.318	4.628	< 0.001
Thyrotoxicosis										
Without	45	2373.77	1895.72	92	9799.05	938.87	2.875	1.930	3.852	< 0.001
With	4	178.96	2235.14	5	553.91	902.67	3.527	2.367	4.724	< 0.001
Septicemia										
Without	47	2515.05	1868.75	94	10,190.66	922.41	2.885	1.936	3.865	< 0.001
With	2	37.68	5307.86	3	162.30	1848.43	4.089	2.745	5.478	< 0.001
Pneumonia										
Without	42	2260.70	1857.83	88	9407.45	935.43	2.829	1.898	3.789	< 0.001
With	7	292.03	2397.01	9	945.51	951.87	3.586	2.407	4.804	< 0.001
CLD										
Without	41	2213.56	1852.22	87	9388.34	926.68	2.847	1.911	3.813	< 0.001
With	8	339.17	2358.70	10	964.62	1036.68	3.240	2.174	4.341	< 0.001
Injury										
Without	37	1987.55	1861.59	85	9111.41	932.90	2.842	1.907	3.807	< 0.001
With	12	565.18	2123.22	12	1241.55	966.53	3.129	2.099	4.191	< 0.001
Tumor										
Without	47	2477.40	1897.15	94	10,056.88	934.68	2.891	1.940	3.872	< 0.001
With	2	75.33	2654.98	3	296.08	1013.24	3.732	2.505	4.999	< 0.001
Season										
Spring	10	584.12	1711.98	22	2435.31	903.38	2.698	1.811	3.615	< 0.001
Summer	12	687.03	1746.65	24	2655.14	903.91	2.752	1.847	3.686	< 0.001
Autumn	13	649.85	2000.46	24	2588.67	927.12	3.073	2.063	4.117	< 0.001
Winter	14	631.73	2216.14	27	2673.84	1009.78	3.126	2.097	4.187	< 0.001
Urbanization level										
1 (The highest)	17	734.37	2314.91	31	2912.68	1064.31	3.097	2.079	4.149	< 0.001
2	15	781.59	1919.16	28	3123.44	896.45	3.049	2.046	4.084	< 0.001
3	8	423.65	1888.35	19	2139.70	887.97	3.028	2.032	4.057	< 0.001
4 (The lowest)	9	613.12	1467.90	19	2177.14	872.70	2.395	1.608	3.209	< 0.001
Level of care										
Hospital center	19	932.56	2037.40	28	2960.75	945.71	3.068	2.059	4.110	< 0.001
Regional hospital	17	876.21	1940.17	38	4020.84	945.08	2.923	1.962	3.916	< 0.001
Local hospital	13	743.96	1747.41	31	3371.37	919.51	2.706	1.816	3.625	< 0.001

PYs = Person-years; aHR = Adjusted Hazard ratio: Adjusted for the variables listed in Table 3.; CI = confidence interval, DM: diabetes mellitus; HTN: hypertension; CLD: chronic liver disease.

**Table 5 healthcare-11-01462-t005:** Factors of poor prognosis subgroups by using Cox regression and Bonferroni correction for multiple comparisons.

	Visual Loss	With			Without *(Reference)*			With vs. Without *(Reference)*			
Sensitivity Test	Poor Prognosis Subgroups	Events	PYs	Rate (per 10^5^ PYs)	Events	PYs	Rate (per 10^5^ PYs)	aHR	95% CI	95% CI	*p*
Overall	Overall	49	2552.73	1919.51	97	10,352.96	936.93	2.956	1.984	3.960	<0.001
	Mental disorders	40	2552.73	1566.95	75	10,352.96	724.43	3.087	2.073	4.143	<0.001
	Anxiety	10	2552.73	391.74	16	10,352.96	154.55	3.715	2.481	4.969	<0.001
	Depression	12	2552.73	470.08	18	10,352.96	173.86	3.859	2.587	5.253	<0.001
	Bipolar	3	2552.73	117.52	9	10,352.96	86.93	1.930	1.123	2.648	0.003
	Sleep disorders	7	2552.73	274.22	11	10,352.96	106.25	3.584	2.460	4.709	<0.001
	PTSD/ASD	2	2552.73	78.35	2	10,352.96	19.32	5.835	3.866	7.892	<0.001
	Dementia	1	2552.73	39.17	2	10,352.96	19.32	2.879	1.950	3.926	<0.001
	Eating disorders	0	2552.73	0.00	2	10,352.96	19.32	0.000	-	-	0.992
	SRD	1	2552.73	39.17	3	10,352.96	28.98	1.604	1.077	2.586	0.019
	Psychotic disorders	2	2552.73	78.35	5	10,352.96	48.30	2.276	1.557	3.244	<0.001
	Autism	0	2552.73	0.00	2	10,352.96	19.32	0.000	-	-	0.987
	Other mental disorders	2	2552.73	78.35	5	10,352.96	48.30	2.186	1.422	3.097	<0.001
	Suicide	1	2552.73	39.17	4	10,352.96	38.64	1.436	0.976	1.946	0.104
	All-caused mortality	8	2552.73	313.39	18	10,352.96	173.86	2.578	1.716	3.444	<0.001
	Suicide mortality	1	2552.73	39.17	3	10,352.96	28.98	2.001	1.359	2.663	<0.001
	Non-suicide mortality	7	2552.73	274.22	15	10,352.96	144.89	2.682	1.784	3.606	<0.001
In the first year excluded	Overall	40	2390.11	1673.56	76	9690.47	784.28	3.709	2.066	4.128	<0.001
	Mental disorders	32	2390.11	1338.85	58	9690.47	598.53	3.192	2.145	4.276	<0.001
	Anxiety	7	2390.11	292.87	12	9690.47	123.83	3.373	2.270	4.522	<0.001
	Depression	10	2390.11	418.39	13	9690.47	134.15	4.453	2.986	5.963	<0.001
	Bipolar	2	2390.11	83.68	7	9690.47	72.24	1.654	1.113	2.207	<0.001
	Sleep disorders	6	2390.11	251.03	9	9690.47	92.87	3.935	2.580	5.174	<0.001
	PTSD/ASD	2	2390.11	83.68	2	9690.47	20.64	5.526	3.884	7.801	<0.001
	Dementia	1	2390.11	41.84	2	9690.47	20.64	2.270	1.943	3.868	<0.001
	Eating disorders	0	2390.11	0.00	2	9690.47	20.64	0.000	-	-	0.999
	SRD	1	2390.11	41.84	1	9690.47	10.32	5.745	3.882	7.759	<0.001
	Psychotic disorders	2	2390.11	83.68	4	9690.47	41.28	2.893	1.940	3.894	<0.001
	Autism	0	2390.11	0.00	2	9690.47	20.64	0.000	-	-	0.998
	Other mental disorders	1	2390.11	41.84	4	9690.47	41.28	1.448	0.824	1.935	0.093
	Suicide	1	2390.11	41.84	4	9690.47	41.28	1.493	0.925	1.943	0.075
	All-caused mortality	7	2390.11	292.87	14	9690.47	144.47	2.898	1.942	3.875	<0.001
	Suicide mortality	1	2552.73	39.17	3	10,352.96	28.98	2.402	1.362	3.780	<0.001
	Non-suicide mortality	6	2552.73	235.04	11	10,352.96	106.25	3.765	2.087	7.184	<0.001
In the first 5 years excluded	Overall	24	1751.06	1370.60	48	7136.88	672.56	2.941	1.973	3.945	<0.001
	Mental disorders	19	1751.06	1085.06	37	7136.88	518.43	2.982	2.008	4.001	<0.001
	Anxiety	5	1751.06	285.54	8	7136.88	112.09	3.637	2.436	4.874	<0.001
	Depression	8	1751.06	456.87	9	7136.88	126.11	5.219	3.467	6.920	<0.001
	Bipolar	2	1751.06	114.22	4	7136.88	56.05	2.878	1.911	3.892	<0.001
	Sleep disorders	3	1751.06	171.32	6	7136.88	84.07	2.908	1.942	3.903	<0.001
	PTSD/ASD	0	1751.06	0.00	1	7136.88	14.01	0.000	-	-	0.987
	Dementia	0	1751.06	0.00	1	7136.88	14.01	0.000	-	-	0.979
	Eating disorders	0	1751.06	0.00	1	7136.88	14.01	0.000	-	-	0.999
	SRD	0	1751.06	0.00	1	7136.88	14.01	0.000	-	-	0.993
	Psychotic disorders	1	1751.06	57.11	2	7136.88	28.02	2.920	1.976	3.923	<0.001
	Autism	0	1751.06	0.00	1	7136.88	14.01	0.000	-	-	0.999
	Other mental disorders	0	1751.06	0.00	3	7136.88	42.04	0.000	-	-	0.996
	Suicide	0	1751.06	0.00	2	7136.88	28.02	0.000	-	-	0.993
	All-caused mortality	5	1751.06	285.54	9	7136.88	126.11	3.231	2.169	4.339	<0.001
	Suicide mortality	0	2552.73	0.00	2	10,352.96	19.32	0.000	-	-	0.992
	Non-suicide mortality	5	2552.73	195.87	7	10,352.96	67.61	3.681	2.304	6.724	<0.001

PYs = Person-years; aHR = Adjusted Hazard ratio: adjusted for the variables listed in Table 3. CI = confidence interval, DM: diabetes mellitus; HTN: hypertension; CLD: chronic liver disease.

## Data Availability

Restrictions apply to the availability of these data. Data were obtained from National Health Insurance database and are available from the authors with the permission of National Health Insurance Administration of Taiwan.

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
