# Peer review of "A Population-Based Cohort Study of the Association between Visual Loss and Risk of Suicide and Mental Illness in Taiwan"

_healthcare, 2023, doi:10.3390/healthcare11101462_

Round 1

Reviewer 1 Report

Introduction

The introduction seems adequate for the paper. It provides a comprehensive overview of the prevalence of visual impairment and its psychosocial and health consequences, as well as the previous research on the association between visual impairment and suicide. The introduction also highlights the need for population-based studies on this topic and explains the rationale for conducting the study in a representative Taiwanese population.

However, why do the authors state that the prevalence of suicide is low? The suicide remains a problem of great magnitude worldwide, with more than 800,000 deaths annually (WHO, 2021), it is one of the leading causes of external death in young people.

The papers used to support the claims about suicide are outdated.

Materials and Methods

I recommend to start this section with a statement about the type of methodology (Study design) used in the study.

Data sources

The information provided is sufficient for describing the data source used in the study.

Participants

It is a suitable description of the participants section in the study. It outlines the inclusion and exclusion criteria for the study, as well as the categories of visual impairment according to the ICD. It also explains how the control group was selected and matched to the study cohort. Overall, it provides clear and concise information about the study participants.

Outcome measurement and comorbidities

This section provides a properly description of the variables that were evaluated, encompassing medical comorbidities, risk factors for poor prognosis, and demographic variables. Moreover, the text offers explicit information regarding the definitions and criteria utilized for each variable.

Statistical analyses

It provides clear information on the variables assessed, the statistical tests used, and the significance level. Additionally, it explains the primary goal of the study and the methods used to determine the risk of poor prognosis. The use of specific statistical software and correction for multiple comparisons is also noted, which adds to the rigor of the analysis.

To improve this section:

·         Introduce the section with a brief explanation of the purpose of the statistical analyses and how they were conducted. For example: "To investigate the association between visual loss and risk of suicide and mental illness, we conducted various statistical analyses to compare the clinical characteristics of the participants in the case and control groups”

·         Divide the section into different paragraphs that correspond to the different types of statistical analyses used (e.g., "Comparison of categorical variables," "Comparison of continuous variables," "Cox regression analysis," "Kaplan-Meier analysis," etc.). This will help readers to navigate the section more easily and understand the different analyses that were performed.

Minor

Spelling mistakes: at the beginning of the first paragraph, a T is missing.

Results

The results are well presented and structured in the corresponding tables.

Discussion

The disussion is well presented. It presents evidence from various studies to support the association between VI and poor mental health outcomes, providing a comprehensive overview of the existing literature. However, some refenrences are outdated.

Additionally, the discussion recognizes the necessity for more studies to explore the occurrence and effects of PTSD among individuals with VI, underscoring a deficiency in existing literature and pointing towards avenues for future research. It also stresses the significance of ophthalmologists in identifying and addressing mental health issues in this population, an essential measure towards enhancing their general welfare.

The study has a few limitations that should be acknowledged. Firstly, the study's design is a population-based cohort study, which can establish an association between VI and mental health outcomes but does not establish a cause-and-effect relationship. Therefore, the conclusion should have been tempered with caution and emphasized the need for further research to explore the causal mechanisms between VI and mental health outcomes. Secondly, the results of some analyses are not statistically significant, which may impact the credibility of the findings. Lastly, the discussion overlooks the discussion of potential confounding factors, such as socioeconomic status, social support, and access to healthcare, that may affect the association between VI and poor mental health outcomes.

Despite these limitations, the study provides valuable insights into the mental health needs of people with VI and emphasizes the importance of addressing their mental health needs. The recommendation for future studies with higher methodological rigor is appropriate to address the limitations of the current study and to strengthen the evidence base on this important topic.

Reviewer 2 Report

Thank you for the opportunity to review this very interesting and necessary study. The findings are very-well presented, however more work is needed to clearly explain the WHY of this study. Further comments are attached
